# Deep Learning and Artificial Intelligence Applied to Model Speech and Language in Parkinson’s Disease

**DOI:** 10.3390/diagnostics13132163

**Published:** 2023-06-25

**Authors:** Daniel Escobar-Grisales, Cristian David Ríos-Urrego, Juan Rafael Orozco-Arroyave

**Affiliations:** 1GITA Lab, Faculty of Engineering, University of Antioquia, Medellín 050010, Colombia; daniel.esobar@udea.edu.co (D.E.-G.); cdavid.rios@udea.edu.co (C.D.R.-U.); 2LME Lab, University of Erlangen, 91054 Erlangen, Germany

**Keywords:** Parkinson’s disease, natural language processing, speech processing, convolutional neural networks, Wav2Vec, word embeddings

## Abstract

Parkinson’s disease (PD) is the second most prevalent neurodegenerative disorder in the world, and it is characterized by the production of different motor and non-motor symptoms which negatively affect speech and language production. For decades, the research community has been working on methodologies to automatically model these biomarkers to detect and monitor the disease; however, although speech impairments have been widely explored, language remains underexplored despite being a valuable source of information, especially to assess cognitive impairments associated with non-motor symptoms. This study proposes the automatic assessment of PD patients using different methodologies to model speech and language biomarkers. One-dimensional and two-dimensional convolutional neural networks (CNNs), along with pre-trained models such as Wav2Vec 2.0, BERT, and BETO, were considered to classify PD patients vs. Healthy Control (HC) subjects. The first approach consisted of modeling speech and language independently. Then, the best representations from each modality were combined following early, joint, and late fusion strategies. The results show that the speech modality yielded an accuracy of up to 88%, thus outperforming all language representations, including the multi-modal approach. These results suggest that speech representations better discriminate PD patients and HC subjects than language representations. When analyzing the fusion strategies, we observed that changes in the time span of the multi-modal representation could produce a significant loss of information in the speech modality, which was likely linked to a decrease in accuracy in the multi-modal experiments. Further experiments are necessary to validate this claim with other fusion methods using different time spans.

## 1. Introduction

### 1.1. Motivation

Neurodegenerative diseases are the leading cause of disability and the second cause of death worldwide. Parkinson’s disease (PD) is the second most prevalent neurodegenerative disorder worldwide after Alzheimer’s disease [1]. PD is characterized by different motor and non-motor symptoms, such as bradykinesia, rigidity, cognitive decline, sleep disorders, and others [2].

About 90% of PD patients develop speech disorders [3], which makes speech analysis a very good biomarker to assess and monitor the disorder. The speech symptoms suffered by PD patients are known as hypokinetic dysarthria. This type of dysarthria appears as the result of losing movement control of the muscles and limbs involved in the speech production process. It affects different dimensions of speech production, including respiratory, phonatory, articulatory, and prosodic dimensions [4,5]. All these factors also affect the intelligibility of patients, as well as limit their quality of life and effective communication [6,7].

Regarding non-motor symptoms developed by PD patients, they range from mood disorders to depression. These symptoms can be modeled using the information of language [8], facial expressions [9], and others. This study focuses on modeling speech and language to evaluate motor and non-motor aspects that are affected due to PD. Although few studies consider language in PD, recent advances have shown that different cognitive impairments, which are reflected in language production, are associated with PD [10]. Similarly, different neurological analyses show that action verb processing depends on motor brain circuits, including the basal ganglia. Therefore, the damage of these regions or networks has been hypothesized to yield differential deficits in processing action verbs [11,12]. Additionally, deficits in verb inflection [13], verbal fluency [14], and verb generation, have been studied in PD [15,16]. In addition, other scientists have connected language deficits to working memory and executive function [17]. There is also evidence showing that syntaxis is also affected in PD [18].

### 1.2. Literature Review in Speech Analysis for PD Evaluation

The scientific community has addressed different approaches based on speech signals and deep learning architectures. In 2019, two frameworks based on CNNs to classify PD using sets of vocal features were proposed in [19]. The first one combined different feature sets before feeding the CNN (feature-level fusion), and the second one passed the feature sets through the parallel input layers, which were directly connected to convolution layers (model-level fusion). The frameworks were trained and evaluated with the UCI Machine Learning repository database, which consists of vocal features extracted from sustained phonations of the vowel /A/ produced by 188 PD patients and 64 HC subjects. Experimental results showed that the second framework worked best, where it yielded an accuracy of 86.9%. Later, in [20], the authors proposed a methodology based on different pre-trained convolutional neural networks (CNNs) such as ResNet-18, ResNet-50, and ResNet-101. A long short-term (LSTM) network was also evaluated. All architectures were fed with mel spectrograms obtained from the speech signals of a corpus called PC-GITA [21], which were collected from a total of 100 speakers (50 of them with PD). The highest classification performance reported by the authors was obtained from the ResNet-101 + LSTM model, with an accuracy of 98.61%. Unfortunately, the results were over-optimistic, because the hyper-parameters were optimized on the test set. Speaker independence was not declared in the paper. Additionally, in 2021, the authors in [22] used bidirectional LSTMs and followed the paradigm originally proposed in [23] to model the energy in the onset and offset transitions and to encode relevant articulatory information in dysarthric speech. The study considered recordings collected at the GYENNO SCIENCE PD Research Center (https://www.gyenno.com/aboutUs/introduce-en (accessed on 21 June 2023)), which include 15 HC subjects and 30 PD patients. The authors showed that accuracy was improved compared to classical approaches. Recently, the same authors presented a study in [24] and considered recordings from the PC-GITA database, together with the ones collected in the GYENNO SCIENCE PD Research Center. The authors implemented a model that combined 1D-CNN and 2D-CNN to capture time and frequency information, and they reported accuracies of 92% and 81% for each database, respectively.

### 1.3. Literature Review in Language Analysis for PD Evaluation

Language has been less studied in PD patients; however, some works have reported different patterns associated with PD. About seven years ago, in [15], the authors evaluated the impact of PD on spontaneous discourses produced by a group of 50 PD patients and 50 HC subjects. Three feature sets were considered: semantic fields (via latent semantic analyses), grammatical (using parts-of-speech tagging), and word-level repetitions (with graph embedding tools). The grammatical features achieved the best classification results, with accuracies of 75% using a KNN classifier. In addition, a linear model was estimated using the word-level repetition features. The authors reported a Pearson’s correlation coefficient of 0.77 between the predicted scores and the MDS-UPDRS-III scores. Four years later, in [18], the authors extracted features such as the morpheme frequencies in a specific category, morpheme lengths, and others. Morphological features were used to model language produced by speakers of three different languages: Spanish, German, and Czech. The Spanish corpus contained 50 PD patients and 50 HC subjects, the German one had a total of 176 speakers (88 with PD), and the Czech corpus consisted of 36 subjects (20 with PD). All subjects were asked to describe a typical day in their life, and manual transcriptions were generated to proceed with language analyses. The authors trained a support vector machine (SVM) with a linear kernel following a leave-one-out cross-validation (LOOCV) and reported accuracies of 71% to classify between PD patients and HC subjects in Spanish and German. For the Czech cohort, the accuracy was 80%. The authors also estimated the neurological states of the patients and evaluated their models with the Pearson’s correlation coefficient computed between the estimated values and the labels assigned by expert neurologists according to the third section of the Movement Disorders Society - Unified Parkinson’s Disease Rating Scale (MDS-UPDRS-III) score [25]. The highest correlation was found for the Czech data, with *r* = 0.61, while, for Spanish and German data, the results were 0.35 and 0.26, respectively.

On the other hand, deep learning architectures have also been considered for modeling language patterns in patients with PD. In [26], transliterations of the monologue task of the PC-GITA corpus were used to classify PD patients and HC subjects. Popular natural language processing (NLP) methods, such as Word2Vec (W2V) [27], Bag of Words (BoW) [28], and Term Frequency–Inverse Document Frequency (TF–IDF) [29], were used to create numerical representations from transliterations. An SVM was used as a classifier, and accuracies of up to 72% were obtained using the W2V approach. In [30], 1D-CNN and bidirectional LSTMs were proposed to model the linguistic content of PD patients when typing different sentences. First, the authors obtained the numerical representation of each character using an adaptation of the continuous BoW method [27]. The embeddings resulting from each sentence were used as inputs for the 1D-CNN. The approach was tested in Spanish and English, and the results showed AUCs of up to 0.8% and 0.7%, respectively. Recently, the authors in [31] presented the Proximity to Reference Semantic Field (P-RSF) metric, which was used to weight action and non-action concepts included by 80 participants (40 of them with PD) while retelling action and non-action stories. Accuracies of 85% and 43% were obtained with each story, respectively, thus indicating that retelling action stories is a more sensible task to detect PD patients than other tasks where the action/non-action paradigm is not taken into account. A summary with recent studies that consider speech and language analyses for PD detection and evaluation is presented in Table 1.

### 1.4. Contribution of This Study

This work is in line with the interest of the scientific community in evaluating and understanding PD and its related symptoms by means of modeling speech and language biomarkers. The following are the main contributions of this paper:(i)Speech recordings were modeled using different modern methods based on deep learning architectures, including representations extracted from the Wav2Vec 2.0 model, as well as 1-dimensional and 2-dimensional CNNs, which were all originally proposed in this paper.(ii)Transliterations of the recordings were modeled using different strategies to represent language patterns, including models such as W2V, BERT, and BETO. We introduced an original method based on CNNs adapted to NLP to consider different n-gram relationships/contexts among the words.(iii)The best representations from each modality were combined using three different fusion strategies; namely, these were early, joint, and late fusion.

The rest of the paper is structured as follows: Section 2 describes the corpora and the methods used in the study. Section 3 presents the experiments and results, and, finally, Section 4 contains the conclusions and future work.

## 2. Materials and Methods

### 2.1. Data

The database considered in this study includes 165 Colombian Spanish native speakers, wherein 80 of them suffer from PD. All speakers were matched in age and gender, and each participant was asked to describe a regular day in his/her life for approximately 90 s. Speech recordings were collected at a sampling frequency of 44.1 kHz and 16-bit resolution. The recordings were normalized using a GSM full-rate compression technique and down-sampled to 8 kHz [34]. All patients were recorded in an on state during the session, i.e., no more than three hours after the medication intake. A neurologist expert evaluated every PD patient to determine the disease severity according to the MDS-UPDRS-III. Table 2 shows the demographic and clinical information of the speakers. Figure 1 shows the distribution of the number of words in the transliterations generated using the Amazon transcribe service for HCs and PD patients. Both distributions followed the same distribution according to the Mann–Whitney test, with a confidence level of α=0.01.

### 2.2. Methods

Figure 2 illustrates a general overview of the methodology proposed in this work. Speech and language information were extracted independently using different characterization strategies. For speech signals, we used the 1D-CNN, 2D-CNN, and a Wav2Vec 2.0 models. For language, we used three state-of-the-art word-embedding models: W2V, BERT, and BETO. These three methods create numerical representations of each word in the transliteration. These embeddings were further processed using a CNN model adapted to NLP. Another approach, based on the estimation of several statistical functionals of the embeddings, was also considered. Finally, the best representations of speech and language were combined following early, joint, and late fusion strategies. Additional details about the methods are presented below.

### 2.3. Speech

Speech embeddings were created using time-frequency representations to feed 1D and 2D CNNs. In addition, the Wav2vec 2.0 approach was also used to generate another speech representation. Details of each method are presented below.

#### 2.3.1. 1D-CNN

This approach is based on 1D convolutional layers followed by LSTMs. The 1D convolutional layer configurations are typically used to model sequential data. The main idea is to extract different representations in the temporal domain. The layer consists of 2 main elements, the number of channels and the kernel, which act as filters for the input data and move across the signal to extract information based on its weight and size. The layer includes as many kernels as channels. Thus, the output data have several representations of the same signal, which correspond to the number of channels. The kernel’s weights are learned during the training process, while its size is specified as a hyperparameter. The output value of a convolutional layer with input size $\left(N,C_{in},L\right)$ and output $\left(N,C_{out},L\right)$ is described in Equation (Equation 1).
(1)$$\mathrm{out}\left(N_{i},C_{out_{j}},L\right)=\mathrm{bias}\left(C_{out_{j}},L\right)+\sum_{k=0}^{C_{in}-1}\mathrm{weight}\left(C_{out_{j}},k,L\right)★\mathrm{input}\left(N_{i},k,L\right),$$
where ★ is the valid cross-correlation operator, *N* is a batch size, *C* denotes a number of channels, and *L* is the length of the signal sequence. Then, each convolutional layer is commonly followed by a temporal max pooling layer, which is similar to a down-sampling technique used to reduce the temporal size of the data.

The sequential information represented by the convolutional and pooling layers is modeled with LSTMs. This network is a type of cell designed to retain sequential information over longer durations. It incorporates a cell state $c_{t}$, which serves to store long-term information. Additionally, the LSTM introduces three concepts: the forget gate $f_{t}$, the input gate $i_{t}$, and the output gate $o_{t}$. To decide what information is forgotten from the cell state, the forget gate $f_{t}$ is used. Equation (Equation 2) defines the $f_{t}$, where $h_{t-1}$ is the previous hidden state, $x_{t}$ is the current input, $W_{f}$ is the weights matrix, and $b_{f}$ is the bias in the forget gate. The relevant information to be added to the cell state using the input gate $i_{t}$ and the vectors of new candidates $\tilde{c}$ are defined by Equations (Equation 3) and (Equation 4), respectively. The cell state is updated using Equation (Equation 5). Finally, the output gate $o_{t}$ determines the new hidden state; this gate is defined by Equation (Equation 6), and the new hidden state $h_{t}$ is defined by Equation (Equation 7), where, * denotes a point-wise (Hadamard) multiplication operator.
(2)$$f_{t}=\sigma \left(W_{f}\left[h_{t-1},x_{t}\right]+b_{f}\right).$$
(3)$$i_{t}=\sigma \left(W_{i}\left[h_{t-1},x_{t}\right]+b_{i}\right).$$
(4)$$\tilde{c}=tanh\left(W_{c}\left[h_{t-1},x_{t}\right]+b_{c}\right).$$
(5)$$c_{t}=f_{t}*c_{t-1}+i_{t}*\tilde{c},$$
(6)$$o_{t}=\sigma \left(W_{o}\left[h_{t-1},x_{t}\right]+b_{o}\right).$$
(7)$$h_{t}=o_{t}*tanh\left(c_{t}\right).$$

In our particular case, the architecture was composed of two 1D convolutional layers with 16 and 32 channels, respectively. Each layer was followed by a temporal max pooling with a kernel size of 2. The characterization performed by the convolutional layers was the input into an LSTM responsible for performing the temporal analysis of the network. This stage was composed of 2 LSTM layers with 64 cells. Finally, the output of the LSTM fed a fully connected network to make the final decision about whether a given speech recording belonged to a PD patient or a healthy speaker.

#### 2.3.2. 2D-CNN

The 2D-CNN has been widely used in computer vision applications [35], and, recently, its application has been extended to other domains such as speech processing and natural language processing. Equation (Equation 8) defines the convolutional layer, where I is an image with only one channel, H is a matrix known as a kernel and whose size must be smaller than the image dimension, *G* is the output of the convolutional layer and is known as the feature map, *m* and *n* are the indexes of the rows and columns of *G*, respectively, and * represents the convolution operator. This expression can be extended for images with different channel numbers. The convolutional layer is followed by a pooling layer, which is used to reduce the feature map size and speed up calculations. In this layer, the feature map is divided into different regions, and then a simple operation is performed. For instance, the maximum value from each region used to build a new feature map of the whole dimension is smaller than the original feature map. Finally, the flattening layer is used to obtain a one-dimensional array from a multi-dimensional array, which allows for creating a valid vector that feeds a fully connected layer that is responsible for performing the classification.
(8)$$G\left[m,n\right]=\left(I*H\right)\left[m,n\right]=\sum_{j}^{}\sum_{k}^{}H\left[j,k\right]I\left[m-j,n-k\right].$$

One of the topologies that has excelled in CNN is the ResNet one. ResNet allows for continuous learning of the gradient by adding additional information to the output of each block through the addition operation. In a plain CNN topology, consecutive layers use a nonlinear mapping function. In contrast, ResNet includes an identity mapping where the block input is added to the output. ResNet defines a residual function F(x)=H(x)−x, where F(x) represents the stacked layers of the neural network, H(x) represents the direct mapping of *x*, and *x* is the identity function. ResNet architecture aims for the residuals to converge to zero (F(x)=0) rather than adjusting an identity mapping (*x*, input = output) through the network layers. It suggests that finding a solution such as F(x)=0 is easier than achieving F(x)=x when using the stack of nonlinear CNN layers as a function.

Particularly, the second approach is one of the most used in state-of-the-art approaches, and it is based on 2D convolution layers, which are fed by time–frequency representations. First, each recording was segmented into 500 ms chunks with 250 ms time shifts. Next, each chunk was transformed into a time–frequency representation using short-time Fourier transform (STFT). Then, each representation was transformed into a mel scale spectrogram with 128 mel filters; therefore, the final representation to feed the CNN was 128 × 63. The CNN architecture was based on a ResNet topology; specifically, the architecture was composed of a convolutional layer in its input with 16 channels to adjust the spectrograms to a ResNet topology; as a result, the architecture had 6 residual blocks and 3 main blocks with 16, 32, and 64 feature maps. Then, we performed an average pooling per channel representing each chunk in an embedding of 64 dimensions. Finally, we used a fully connected network for the classification to make the final decision [36].

#### 2.3.3. Wav2vec 2.0

The third approach was based on the Wav2vec 2.0 architecture, which is based on transformers [37]. The main idea of this architecture is to encode speech audio via a multi-layer convolutional neural network and then mask spans of the resulting latent speech representations. The latent representations are fed to a transformer network to build contextualized representations. The model is trained via a contrastive task, where the true latent is to be distinguished from distractors. Particularly in this work, we used a pre-trained Wav2Vec 2.0 model available in Pytorch to obtain a speech representation for each recording from scratch in its input. This architecture was pre-trained on 960 h of unlabeled data from the LibriSpeech dataset and fine-tuned for ASR on the same audio files with the corresponding transcripts. After that, we performed a temporal mean to obtain a 768-dimensional representation per audio file. Finally, we used a fully connected layer to decide whether the recording corresponded to a PD patient or HC subject.

### 2.4. Language

Three state-of-the-art word-embedding models were used to obtain the numerical representations of the words in the transliterations: W2V, BERT, and BETO. These embeddings allow for obtaining an embedding matrix $M^{n\times d}$ per transliteration, where *n* is the number of words in the transliteration, and *d* is the dimension of the word embedding. Details about each method are presented below.

#### 2.4.1. W2V

This method generates context-independent word embeddings, because the representation of each word is the same regardless of its context. This embedding has been widely used in different NLP tasks, including emotion recognition, depression detection, and others. Furthermore, these numerical representations of the words preserve several semantic properties; for instance, word vectors live in the same vector space such that words sharing a common context in the corpus are geometrically close to each other [38]. W2V representations are obtained using a simple architecture with one hidden layer. This architecture is trained in a self-supervised way; therefore, the architectures can be fine-tuned using unlabeled information. We used a W2V model trained with the Spanish WikiCorpus, which contains 120 million words [39]. The word-embedding dimension was 300, and the model was trained using a skip-gram strategy with eight context words.

#### 2.4.2. BERT

This is one of the most popular context-dependent embeddings; in this case, the representation of the word is not unique, because the representation depends on the word’s context. This word embedding is computed using a transformer architecture’s encoder, which was created initially for machine translation [40]. The most important part of the encoder is the multi-head attention mechanism, which consists of several attention layers running in parallel. Each layer learns contextual relations among words. Similarly to the W2V model, the BERT is trained in a self-supervised way using two strategies: the masked language modeling (MLM) and next sentence prediction (NSP). In this work, we used BERT-Base, a multi-lingual uncased pre-trained model, which was trained with the Multi-Genre Natural Language Inference (MultiNLI) corpus.

#### 2.4.3. BETO

This word-embedding model is a Spanish version of BERT proposed in [41]. The model was trained with Spanish data from Wikipedia and all of the resources of the OPUS project [42]. The source code to compute the BERT and BETO embeddings is available online (https://github.com/PauPerezT/WEBERT (accessed on 21 June 2023) [43].

The aforementioned embeddings were used to create an embedding matrix for each transliteration, which was processed following two approaches: (i) A static representation was created per subject by computing four statistical functionals—mean, standard deviation, skewness, and kurtosis. This static representation was used to train an SVM and make the final decision. (ii) The embedding matrix was used to feed a CNN with different kernel sizes to map different *n*-gram relations between words. The convolution was computed in one dimension, and, in this case, we used three kernel sizes, 2×d, 3×d, and 4×d, to map bi-gram, tri-gram, and four-gram relations, respectively. A max-pooling layer was applied to the output of each filter. All outputs were used to feed a fully connected layer, which made the final decision after a Softmax activation function. Further details about this architecture are presented in [44].

### 2.5. Fusion Strategies

Different fusion strategies were used to combine the best representations of speech and language. Figure 3 shows the different fusion strategies considered in this paper, and additional details are presented below.

#### 2.5.1. Early Fusion

Early fusion is a traditional way of fusing multiple data. This method is known as input-level fusion and refers to merging multiple input modalities into a single feature vector/matrix before training and testing. Input modalities can be merged in many ways, including concatenation, pooling, or applying a gated unit. This strategy has two disadvantages: First, a large amount of information is lost by representing each modality from a static feature vector. Second, it is required to synchronize the time stamp of the different modalities, which potentially results in a loss of information. In this approach, we considered the best representation of each modality (speech and language) to concatenate them.

#### 2.5.2. Joint Fusion

Joint fusion (or intermediate fusion) is the process of joining learned feature representations from the intermediate layers of neural networks with features from other modalities as input into a new model. The main difference with early fusion is that, in joint fusion, the loss is back-propagated to the feature extraction stage of the neural network, thereby generating a loop that enables it to find better feature representations based on multi-modal information. Different modalities can be fused simultaneously into a single shared representation layer or performed gradually using one or several modalities simultaneously. For the joint fusion between speech and language, it was necessary to generate a static representation from the speech model. To do that, we created GMM supervectors encoding dynamic information per speaker. Then, we incorporated the language model in an intermediate stage of the network to be merged with each static speech representation created per participant. Language and speech representations were concatenated prior to employing the fully connected layer.

#### 2.5.3. Late Fusion

Late fusion uses data sources independently, followed by a fusion step at the classification stage. This technique is much simpler than the previous fusion method, particularly when the data sources differ significantly from each other in terms of the sampling frequency and dimensionality. Late fusion offers potentially better performance, because the errors from various models are processed independently; therefore, the errors are assumed to be uncorrelated. There are different rules to determine the optimal way to combine the models. Bayes rules, max fusion, and averag fusion are among the most popular. In this strategy, we obtained the scores of the best representations for each modality; they were then merged and used to train and test an SVM to make the final decision based on this new representation.

## 3. Results

We developed three main experiments in this work: (i) Speech recordings were processed considering the three methods described in Section 2.3. (ii) Automatic transcriptions were analyzed using the three different word-embedding models presented in Section 2.4. (iii) Individual models with the best accuracy for each modality were used to set up a multimodal approach, where both modalities were combined using the early, joint, and late fusion strategies described in Section 2.5. In all experiments, the models were trained and evaluated following the same speaker-independent 10-fold cross-validation strategy. In experiments where an SVM was implemented, we used a Gaussian kernel, and its hyper-parameters were optimized using a grid search with C∈{0.001,0.01,⋯,100} and γ∈{0.0001,0.001,⋯,100}. Finally, for the deep learning methods, the architectures were trained with 200 epochs using the cross-entropy loss function and using an Adam optimizer [45]. We used different regularization strategies, including early stopping with a patience of 40. In addition, dropout and $l_{2}$ regularization were also implemented with values defined between [0.2,0.4,0.6] and [0.0001,0.005,0.001], respectively.

### 3.1. Speech

The results obtained with the three implemented methods are reported in Table 3. Notice that, although, for the Wav2vec 2.0 experiments, we evaluated different segmentation lengths, (1 sec, 2 sec, 5 sec, and the full recording), we only reported the best result, which was obtained with 2 s. The best result was obtained with Wav2vec 2.0, with an accuracy of 88.5% and an F1-score of 88.3%. Similar results were obtained with the 2D-CNN approach, which achieved an accuracy of 84.4%. For the case of the 1D-CNN approach, the network seemed to be overfitted towards the negative class (HC subjects), because this approach yielded an accuracy of 72.6% with a high specificity. These results show that obtaining the representations of a robust model such as Wav2vec 2.0 trained with a large amount of data for speech recognition provides better results than training models from scratch, such as those obtained with the 1-D and 2-D CNN introduced in this paper. In addition, more data are needed to obtain more robust and stable results.

Figure 4 shows the histograms and the fitted probability density distributions of the scores obtained when classifying the samples with the best model (Wav2vec 2.0). It can be observed that the error for the discrimination of HC subjects was small (i.e., high specificity), while the error in discriminating PD patients was larger (i.e., relatively low sensitivity). In addition, it is important to mention that, for both classes, the highest bin was located at the extremes of the distribution, which suggests that many patients and controls were correctly classified with a high certainty.

### 3.2. Language

In the language modality, three word-embedding models (W2V, BERT, and BETO) were used to generate the embedding matrix for each transliteration. The resultant embedding matrix for each embedding model was processed using two approaches: statistical functionals and CNNs. Table 4 shows the results obtained in classifying PD patients vs HC subjects for both approaches using the different word-embedding models. The BETO and CNN model achieved an accuracy of up to 77.9%, which outperformed the model based on the BERT and CNN model by about 3.7% and by 4.1% for the model based on the W2V and CNN model. This result was expected, because the BETO was only trained with Spanish data, while the BERT is a multi-lingual model trained with corpora from 104 different languages. Similarly, W2V is a simple word-embedding model and does not incorporate information about the context.

The proposed method based on CNNs systematically outperformed the models based on statistical functionals. According to our observations, this result is because the method modeled the relation between word representations through convolution operations with different contexts using different kernel sizes, which made the approach more robust and suitable to capture paralinguistic information encoded in the language of the patients. Conversely, in the method based on statistical functionals, all word representations were reduced to a single vector, therefore reducing the chances for the classifier to find specific patterns along the encoded texts.

Figure 5 shows the distribution of the scores obtained with the best language model (the BETO and CNN model). In this case, the error for both classes was balanced, as shown in Table 4. Notice also that the area of error was larger than the one obtained with the speech modality using Wav2vec 2.0 (see Figure 4). This is, to some extent, expected, because PD is prominently a motor disorder that eventually (for a subset of patients) results in cognitive impairments, which resultingly affect language production. This is a matter of research, and this is the reason why the subject matter addressed in this paper is relevant, because any finding could guide future directions.

### 3.3. Multi-Modal

Modes that yielded the best results from each modality were combined in the following experiments, i.e., the Wav2vec 2.0 model and the BETO and CNN model. Different resolutions were obtained per modality. The Wav2vec 2.0 approach had different representations per subject—one for each segment of 2 sec. For the case of the BETO and CNN approach, we had one representation per subject. Therefore, we decided to compute a speech representation per subject to perform the combination at a speaker level following three fusion strategies: early, joint, and late fusion. In early fusion, the mean of the embeddings generated per segment was computed to create the speech representation per subject. Then, it was concatenated with the language embedding (resulting from the BETO and CNN model) to form a bi-modal representation, which was used to train an SVM. In the joint fusion, the CNN initially created for language was fine-tuned using information from speech and language. In this case, the speech representation per subject was generated using GMM supervectors, which were created by stacking the means (μ) and the diagonal of the covariance matrix (Σ) of the GMM. We considered two, four, and eight Gaussians to create the static representation, but we only reported the results obtained with eight Gaussians. The GMM supervector per subject was concatenated with the resulting language embedding in an intermediate layer of the CNN, just before the fully connected layer, which developed the classification stage. Finally, in the late fusion strategy, we obtained the speech score per subject by computing the mean of all decision scores per segment. Then, the scores from speech and language were concatenated to form a bi-modal representation with two dimensions, which was used to train and test an SVM.

Table 5 shows the results for the different fusion strategies. The best result was obtained with the joint fusion strategy, where an accuracy of 77.2% was obtained. This accuracy slightly outperformed the one obtained with the late fusion strategy; nevertheless, joint fusion yielded a more balanced sensitivity and specificity. In addition, Figure 6 shows the distribution of the scores resulting from the joint fusion model. Notice that the PD patients and HC subjects were relatively far from the decision threshold. Finally, notice that the standard deviation values, in this case, were smaller than those obtained in previous experiments with language embeddings, thereby indicating that, although there is still work to do, the fusion yielded advantages in robustness and stability.

## 4. Conclusions

This paper presented different methods to automatically discriminate between PD patients and HC subjects using speech and language information. Both modalities were analyzed independently while considering different methods. Finally, the best uni-modal models were used to set up a multi-modal approach, where we aimed to take advantage of joint information in both modalities. Audio recordings were processed using three main approaches: 1D-CNN, 2D-CNN, and Wav2vec 2.0. Transliterations were represented using three word-embedding models: W2V, BERT, and BETO. Static language representations were created by computing statistical functionals. In addition, a novel approach based on a CNN model adapted to NLP was introduced to model language information. Finally, the multi-modal approach consisted of evaluating three fusion strategies where the best individual models were combined: early, joint, and late fusion. To overcome the mismatch between dynamic speech representations and static language representations, speech recordings per speaker were represented via GMM supervectors.

The results indicate that it is possible to discriminate between PD patients and HC subjects with an accuracy of up to 88% using speech modeled with the Wav2vec 2.0 approach, while the accuracy with language was 77% when using the BETO and CNN model. This result validates the fact that PD is mainly a motor disorder, which, therefore, affects speech production mainly and language to a lesser extent. The results obtained with speech modality outperformed the results of the multi-modal approach by up to 11%. However, we showed that the fusion of speech and language information yields more robust and stable results. We hypothesize that this reduced performance is due to the time span in multi-modal representations. We think that the principal loss of information in the speech modality happens when GMM supervectors are created to encode the speaker’s information. Similarly, information on language embeddings is reduced by the computation of mean values. This study has limitations due to the reduced amount of data considered in the experiments, which did not allow us to re-train and fine-tune models such as Wav2Vec or BETO. This constraint prevented us from obtaining representations that were specifically focused on capturing patterns of the disease. In addition, we found that CNNs adapted to NLP yielded the best results in the language modality. We hypothesize that these results could be improved by considering a larger dataset. In future experiments, we plan to implement different language representations to model specific cognitive impairment in PD patients, i.e., incorporating clinically inspired features such as P-RSF [31]. We will also explore diverse data augmentation strategies to fine-tune pre-trained models and generate representations focusing on modeling different disease patterns based on each modality. Finally, we will evaluate different time spans during the fusion stage to mitigate information loss.

## Figures and Tables

**Figure 1 diagnostics-13-02163-f001:**
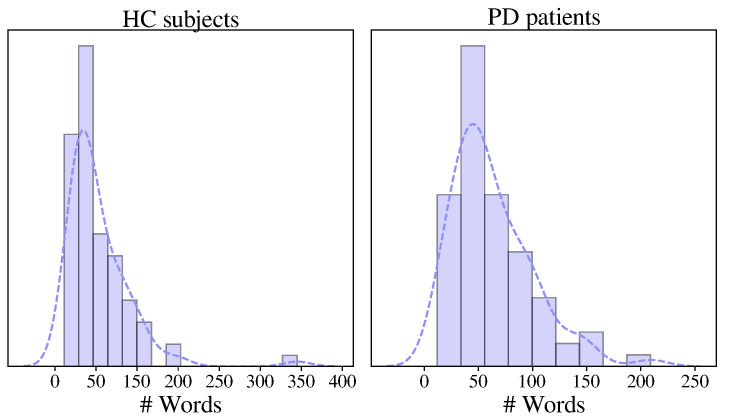
Distribution of the number of words in the transliterations.

**Figure 2 diagnostics-13-02163-f002:**
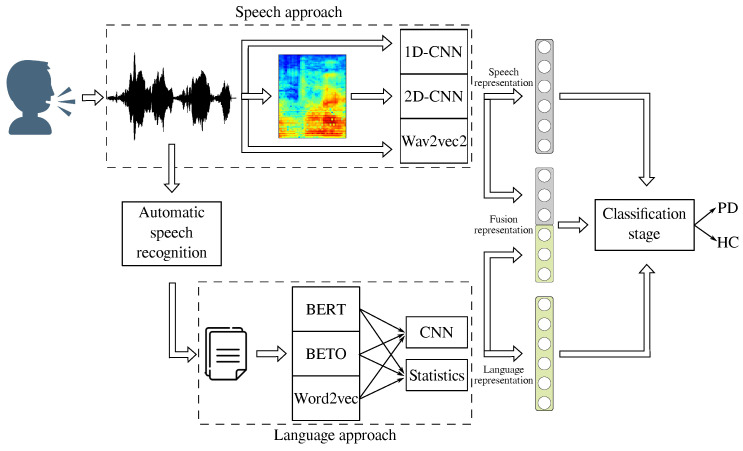
General methodology proposed in this study.

**Figure 3 diagnostics-13-02163-f003:**
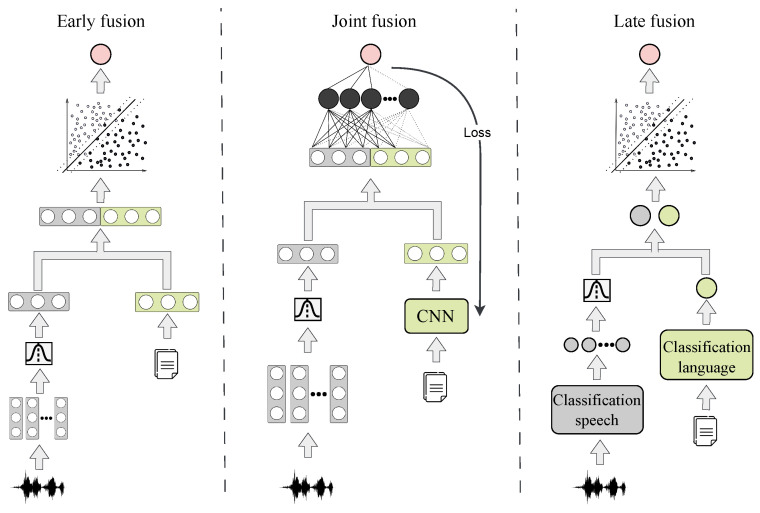
Fusion strategies.

**Figure 4 diagnostics-13-02163-f004:**
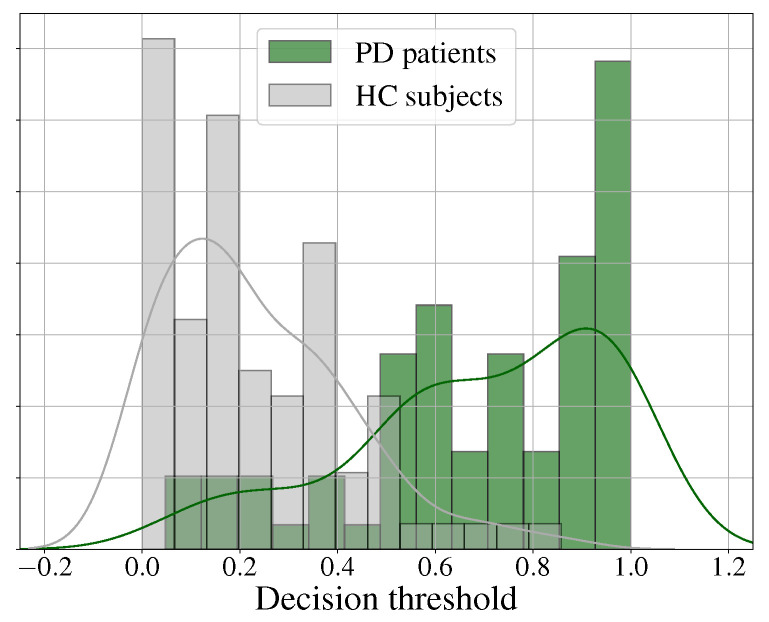
Histogram and the corresponding probability density distribution of the scores obtained from the classification of PD patients and HC subjects in the Wav2vec 2.0 model.

**Figure 5 diagnostics-13-02163-f005:**
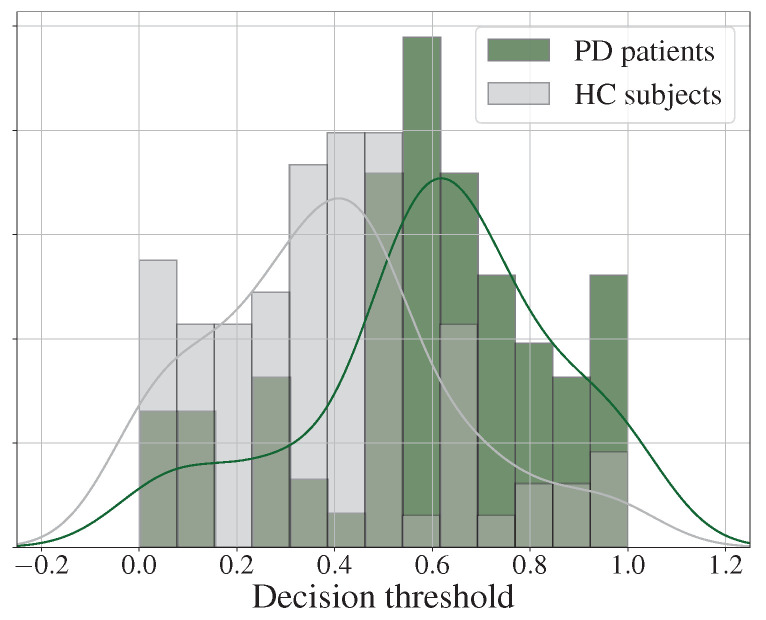
Histograms and probability density distribution of the scores obtained from the classification of PD patients and HC subjects using BETO and CNN.

**Figure 6 diagnostics-13-02163-f006:**
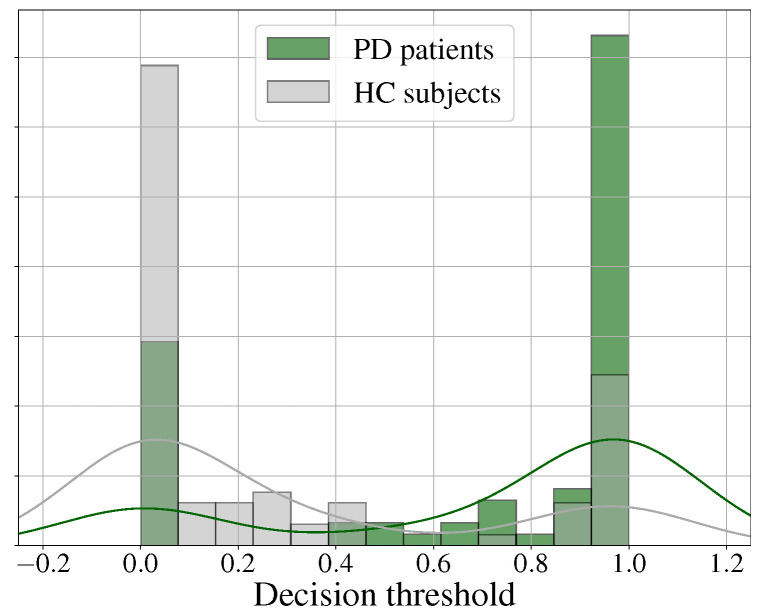
Histogram and the corresponding probability density distribution of the scores obtained from the classification of PD patients and HC subjects in the model with joint fusion.

**Table 1 diagnostics-13-02163-t001:** Summary of recent studies in speech and language analysis for PD detection and evaluation. DB: Database. Acc: Accuracy. AUC: Area under the ROC curve.

Ref.	Datasets	Features/Representation	ML/DL Models	Validation	Results
**Speech analysis**					
[20]	50 PD–50 HC	Mel spectrograms	2D-CNNs and LSTMs	Cross-validation: 10 folds	Acc: 98.6%
[24]	DB1: 30 PD–15 HC DB2: 50 PD–50 HC	Mel spectrograms	1D-CNNs and 2D-CNNs	Training, validation, test split	DB1 Acc: 81.6% DB2 Acc: 92.0%
[22]	30 PD–15 HC	MFCCs and Bark band energies	Bidirectional LSTMs	Cross-validation: 10 folds	Acc: 84.3%
[19]	188 PD–64 HC	Tunable Q-factor and time frequency wavelet transform; MFCCs and vocal fold features	1D-CNNs in parallel	Leave-One-Person-Out cross-validation	Acc: 85.7%
[32]	400 PD–400 HC	Perceptual linear predictive coefficients	Gaussian mixture models	Cross-validation: 5 folds	AUC: 0.95
**Language analysis**					
[31]	40 PD–40 HC	P-RSF metric	SVM	Cross-validation: 10 folds	Acc: 85.0%
[26]	50 PD–50 HC	W2V, BoW TF-IDF	SVM	Cross-validation: 10 folds	Acc: 72.0%
[33]	88 PD–88 HC	Morphological features	SVM	Leave-One-Out cross validation	Acc: 81.0%
[18]	Spanish: 91 PD–57 HC German: 88 PD–88 HC Czech: 20 PD–16 HC	Morphological features	LR SVM SGD	Leave-One-Out cross-validation	Spanish Acc: 71.0% German Acc: 71.0% Czech Acc: 80.0%
[15]	51 PD–50 HC	Semantic fields; Grammatical word-level repetitions	KNN	Leave-One-Out cross-validation	Pearson’s correlation: 0.77
[30]	Spanish: 11 PD–9 HC English: 16 PD–25 HC	Continuous BoW	1D-CNN Bidirectional-LSTM	Cross-validation: 5 folds	Spanish AUC: 0.7 English AUC: 0.8

**Table 2 diagnostics-13-02163-t002:** Clinical and demographic information of the subjects included in this work. [F/M]: Female/Male. Values are reported in terms of mean ± standard deviation.

	PD Patients	HC Subjects	PD vs. HC
Gender [F/M]	38/42	43/42	* *p* = 0.81
Age [F/M]	63.7 ± 7.3/64.5 ± 10.2	60.9 ± 8.2/64.8 ± 10.5	** *p* = 0.38
Range of age [F/M]	51–81/45–86	49–83/42–86	
MDS-UPDRS-III [F/M]	34.6 ± 19.9/38.5 ± 19.6		
Range of MDS-UPDRS-III [F/M]	9–106/7–92		

* *p*—value calculated through Chi-square test. ** *p*—value calculated through *t*-test.

**Table 3 diagnostics-13-02163-t003:** Classification between PD patients vs. HC subjects using speech recordings. Values are reported in terms of mean ± standard deviation.

	2D-CNN	1D-CNN	Wav2vec 2.0
Accuracy	84.4 ± 8.8	72.6 ± 7.9	**88.5 ± 8.3**
Sensitivity	81.3 ± 15.1	53.8 ± 16.8	**82.5 ± 16.9**
Specificity	87.6 ± 15.8	92.5 ± 9.7	**94.0 ± 6.3**
F1-Score	84.3 ± 9.2	68.0 ± 7.7	**88.3 ± 8.7**

**Table 4 diagnostics-13-02163-t004:** Classification between PD patients vs. HC subjects using language embeddings. Values are reported in terms of mean ± standard deviation.

	Statistical Functionals			CNN		
	**W2V**	**BERT**	**BETO**	**W2V**	**BERT**	**BETO**
Accuracy	69.7 ± 6.1	61.8 ± 7.4	62.5 ± 6.9	73.8±10.3	74.2±10.2	**77.9 ± 8.4**
Sensitivity	71.3 ± 17.7	47.5 ± 19.2	56.3 ± 23.9	75.0 ± 22.4	76.3 ± 11.8	**76.4 ± 12.4**
Specificity	68.2 ± 17.4	75.7 ± 22.3	70.0 ± 21.0	72.6 ± 20.2	72.1±17.8	**79.2 ± 15.7**
F1-Score	68.6 ± 8.5	53.1±11.2	56.9 ± 13.3	72.2 ± 14.8	74.2 ± 9.2	**76.9 ± 8.4**

**Table 5 diagnostics-13-02163-t005:** Classification between PD patients vs. HC subjects using three different fusion strategies. Values are reported in terms of mean ± standard deviation.

	Early	Joint	Late
Accuracy	73.9 ± 12.8	**77.2 ± 2.0**	77.6 ± 8.3
Sensitivity	86.3 ± 15.3	**77.3 ± 5.3**	75.0 ± 11.2
Specificity	62.4 ± 26.9	**77.0 ± 4.4**	80.1 ± 11.7
F1-Score	76.5 ± 11.0	**76.3 ± 2.3**	76.4 ± 9.3

## Data Availability

The data considered in this work are not publicly available.

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
