# Peer review of "Deep Learning and Artificial Intelligence Applied to Model Speech and Language in Parkinson’s Disease"

_diagnostics, 2023, doi:10.3390/diagnostics13132163_

Round 1

Reviewer 1 Report (Previous Reviewer 2)

1.Introduction section needs to be re-written to improve its quality and readability.

2.What is the motivation of the proposed work? Research gaps, objectives of the proposed work should be clearly justified

3.Overall, the basic background is not introduced well, where the notations are not illustrated much clear.

4.The literature has to be strongly updated with some relevant and recent papers focused on the fields dealt with in the manuscript.

5.The study lacks a theoretical framework which is important for the reader to grasp the crust of the research.

6. Explain why the current method was selected for the study, its importance and compare with traditional methods.

7.Authors are suggested to include more discussion on the results and also include some explanation regarding the justification to support why the proposed method is better in comparison towards other methods

8.Does this kind of study have never attempted before? Justify this statement and give an appropriate explanation to do so in this paper.

Needs Improvement

Author Response

Reviewer 1

1.Introduction section needs to be re-written to improve its quality and readability.

[Answer] The introduction was re-written, and now it was improved by including more details, more literature review, and a more explicit motivation.

2.What is the motivation of the proposed work? Research gaps, objectives of the proposed work should be clearly justified.

[Answer] The motivation of the study and its contributions have been improved in the Introduction section.

3.Overall, the basic background is not introduced well, where the notations are not illustrated much clear.

[Answer] Methods and background have been improved with mathematical, algorithmic, and concept details.

4.The literature has to be strongly updated with some relevant and recent papers focused on the fields dealt with in the manuscript.

[Answer] Literature review has been improved according to the reviewer’s suggestion.

5.The study lacks a theoretical framework which is important for the reader to grasp the crust of the research.

[Answer] Theoretical and algorithmic details are incorporated into the corrected manuscript.

6. Explain why the current method was selected for the study, its importance and compare with traditional methods.

[Answer] In the case of this manuscript, traditional approaches concern those related to considering speech OR language patterns only. This manuscript not only considers these approaches, but also evaluates different fusion strategies where the question of whether speech and laguage information are complementary is addressed in the topic of Parkinson’s detection.

7.Authors are suggested to include more discussion on the results and also include some explanation regarding the justification to support why the proposed method is better in comparison towards other methods.

[Answer] Results are discussed in Section 3 and further analyzed in the Conclusions. According to our analyses, it is necessary to address further investigations to find language patterns that best model abnormal behavior that appears due to Parkinson’s disease. The main contribution of our study is to highlight the fact that speech and language are complementary, but need to be appropriately modeled to find robust and accurate results.

8.Does this kind of study have never attempted before? Justify this statement and give an appropriate explanation to do so in this paper.

[Answer] The evaluation of different fusion strategies to incorporate speech and language features has not been addressed before. Besides, the work introduces a novel method, based on CNNs, to create language representations (NLP). These contributions are highlighted in Section 1.4. of the corrected manuscript.

Reviewer 2 Report (New Reviewer)

This study examines methodologies to automatically model speech and language 

biomarkers to detect Parkinson’s disease (PD).  CNNs with pre-trained models such as Wav2Vec 2.0, BERT, and BETO  classify PD patients vs. Healthy Control (HC) subjects. 

Speech modality outperforms all language representations.

The paper is well done, in both content and presentation.  I am impressed by the writing, as I generally have to correct many grammatical flaws.

Specific points:

..despite it is a valuable source .. ->

..despite being a valuable source ..

..energy in the onset and offset transitions are shown .. ->

..energy in the onset and offset transitions is shown ..

..Word2Vec (W2V), bag of words (BoW), and Term Frequency-Inverse Document Frequency (TF-IDF) .. - useful to cite these, not just name them

..capture paralingüistics .. - why the umlaut here?

..that area of error is larger .. ->

..that the area of error is larger ..

..approach consisted in evaluating three .. ->

..approach consisted of evaluating three ..

..modeled with Wav2vec 2.0 approach, ..

..modeled with the Wav2vec 2.0 approach, ..

..disorder, therefore, ..

..disorder, and therefore, ..

..prevents us to obtain representations ..

..prevents us from obtaining representations ..

..grant # ES92210001. Also by ..

..grant # ES92210001, and also by ..

The format of the references is strange, e.g., using “De, R.; et al.”

Author Response

Reviewer 2

This study examines methodologies to automatically model speech and language biomarkers to detect Parkinson’s disease (PD). CNNs with pre-trained models such as Wav2Vec 2.0, BERT, and BETO classify PD patients vs. Healthy Control (HC) subjects. Speech modality outperforms all language representations.

The paper is well done, in both content and presentation. I am impressed by the writing, as I generally have to correct many grammatical flaws.

[Answer] We appreciate the reviewer’s evaluation and are really thankful for his/her positive comments.

Specific points:

..despite it is a valuable source .. → ..despite being a valuable source ..

..energy in the onset and offset transitions are shown .. → ..energy in the onset and offset transitions is shown ..

..Word2Vec (W2V), bag of words (BoW), and Term Frequency-Inverse Document Frequency (TF-IDF) .. - useful to cite these, not just name them

..capture paralingüistics .. - why the umlaut here?

..that area of error is larger .. → ..that the area of error is larger ..

..approach consisted in evaluating three .. → ..approach consisted of evaluating three ..

..modeled with Wav2vec 2.0 approach, .. → ..modeled with the Wav2vec 2.0 approach, ..

..disorder, therefore, .. → ..disorder, and therefore, ..

..prevents us to obtain representations .. → ..prevents us from obtaining representations ..

..grant # ES92210001. Also by .. → ..grant # ES92210001, and also by ..

The format of the references is strange, e.g., using “De, R.; et al.”

[Answer] We thank the reviewer for the detailed revision of the manuscript. All comments and corrections have been considered in the corrected version of the manuscript.

This manuscript is a resubmission of an earlier submission. The following is a list of the peer review reports and author responses from that submission.

Round 1

Reviewer 1 Report

The article is very interesting because it considers a new way to confirm the diagnosis of Parkinson's disease. The topic is very timely. In recent years there are several publications that analyse with electronic methods and according to algorithms, the voice, the speech of patients co Parkinson's disease. The authors performed both a speech and a language analysis. Both results are together integrated for a better evaluation. I believe that some things need to be changed before the article can be published. The abstract could be restructured by including purpose, material and methods, results and conclusions. In the introduction, the clinical substrate is too short. There could be more description about the language changes in parkinson disease. In the following paragraphs, articles concerning the analysis of speech and language are commented. A summary table could be useful and reduce the length of the writing. The section on materials and methods is written adequately enough. The conclusion is too short. The limitations of the study are missing.

Reviewer 2 Report

The Research Paper stands Rejected and is NOT RECOMMENDED for Publication as the information highlighted is already known and lots of advanced papers are Published in this regard.

No Proposed methodology or architecture is observed.

No Real Time case study-oriented discussion is there in the paper.

The Novelty and contribution is not encouraging

Reviewer 3 Report

The article "Analysis of speech and language representations for the automatic assessment of Parkinson’s disease" presents the results of machine learning-based research into the detection of Parkinson's disease.
They used different methodologies to model speech and language biomarkers.
1-dimensional and 2-dimensional convolutional neural networks (CNNs), as well as pre-trained models such as Wav2Vec 2.0, BERT and BETO, were used to classify Parkinson's disease (PD) patients and healthy control (HC) subjects.

The overall quality of the paper is sufficient for publication after some minor improvements.

The language and grammar is good, only a few grammatical mistakes can be found in the text. Some of them are listed below for example.

line 16:  time spam
time span

line 19: different time spams
different time spans

lines 142-143: Besides, with introduced an original method based on CNNs adapted to NLP.
Incomplete sentence.

line 349: ...the error for both classes is balanced at is was shown in Table 3.
Bad grammar or typo.

line 367: fine-tunned
fine-tuned

The scientific soundness and description details are close to be acceptable.

Some details are missing for the models and algorithm parameters used.

The presentation of results is clear and convincing, although perhaps little bit inferior to the usual presentation of results in articles presenting practical results of research.

I miss the number of learning session, which would be important for interpreting the statistical results.